# LEVERAGING UNPAIRED DATA FOR VISION-LANGUAGE GENERATIVE MODELS VIA CYCLE CONSISTENCY

**Tianhong Li**[1*] **Sangnie Bhardwaj**[2,3*] **Yonglong Tian**[3] **Han Zhang**[4] **Jarred Barber**[3]
**Dina Katabi**[1] **Guillaume Lajoie**[2] **Huiwen Chang**[5] **Dilip Krishnan**[3]
[1]MIT CSAIL [2]Mila [3]Google Research [4]Google DeepMind [5]OpenAI

## ABSTRACT

Current vision-language generative models rely on expansive corpora of *paired* image-text data to attain optimal performance and generalization capabilities. However, automatically collecting such data (e.g. via large-scale web scraping) leads to low quality and poor image-text correlation, while human annotation is more accurate but requires significant manual effort and expense. We introduce **ITIT** (**In**Te**grating I**mage **T**ext): an innovative training paradigm grounded in the concept of cycle consistency which allows vision-language training on *unpaired* image and text data. ITIT is comprised of a joint image-text encoder with disjoint image and text decoders that enable bidirectional image-to-text and text-to-image generation in a single framework. During training, ITIT leverages a small set of paired image-text data to ensure its output matches the input reasonably well in both directions. Simultaneously, the model is also trained on much larger datasets containing only images or texts. This is achieved by enforcing cycle consistency between the original unpaired samples and the cycle-generated counterparts. For instance, it generates a caption for a given input image and then uses the caption to create an output image, and enforces similarity between the input and output images. Our experiments show that ITIT with unpaired datasets exhibits similar scaling behavior as using high-quality paired data. We demonstrate image generation and captioning performance on par with state-of-the-art text-to-image and image-to-text models with orders of magnitude fewer (only 3M) paired image-text data. Code will be released at https://github.com/LTH14/itit.

## 1 INTRODUCTION

Image-text multimodal training has gained remarkable attention in recent years. Models for text-to-image generation show the impressive ability to synthesize realistic images from textual prompts (Rombach et al., 2022; Chang et al., 2023; Saharia et al., 2022; Yu et al., 2022b; Ramesh et al., 2021). Similarly, image-to-text models have demonstrated advanced image comprehension capabilities by providing precise descriptions of input images (Chen et al., 2023; Wang et al., 2022a; Li et al., 2022; 2023a; Alayrac et al., 2022; Wang et al., 2022b). Training these models to exceptional performance demands datasets comprising hundreds of millions to billions of *paired* image-text samples (Schuhmann et al., 2022). The collection of very large paired datasets comes at a considerable cost as well as concerns of low quality (Jia et al., 2021). On the other hand, diverse and vast unimodal images or texts datasets remain unused in current generative vision-language training (Raffel et al., 2020; Sun et al., 2017; Zhai et al., 2022). This raises a natural question: can we leverage *unpaired* image and text data to facilitate generative vision-language training?

The major problem with using unpaired data during vision-language training is the lack of supervision. Generative vision-language training needs paired text to supervise image-to-text captioning, and paired images to supervise text-to-image generation. To solve this problem, we introduce ITIT, a novel training paradigm that significantly reduces the reliance on paired image-text data, favoring instead the utilization of unpaired image-only and text-only data. As shown in Figure 1, the core principle underlying ITIT is the use of *cycle consistency* losses between cycle-generated images/texts

---

*Co-first authors. This work was done when Tianhong Li interned at and Huiwen Chang worked at Google Research. Correspondence to Tianhong Li <tianhong@mit.edu>.

**Text-Image-Text (T2I2T) Cycle**

Reconstruction Loss

Flamingo | on | a | lake → **T2I** → Synthesized Image → **I2T** → Flamingo | on | a | lake

Unpaired Text $T$ · Reconstructed Text $\hat{T}$

**Image-Text-Image (I2T2I) Cycle**

Reconstruction Loss

Unpaired Image $I$ → **I2T** → A | cat | on | green | grass → **T2I** → Reconstructed Image $\hat{I}$

Synthesized Text

Figure 1: Overview of ITIT. For unpaired data, ITIT first generates the image/text counterpart, and then uses these generated counterparts to reconstruct the original text or image.

and their corresponding original inputs to provide supervision for image-only and text-only data during training.

To enable cycle training, we first unify image-to-text (I2T) and text-to-image (T2I) generation in the same framework, with a bi-directional image-text encoder and disjoint image and text decoders. We tokenize images into discrete visual tokens (Van Den Oord et al., 2017) and combine them with text embeddings from a pre-trained T5 model (Raffel et al., 2020) as input to the joint image-text encoder. For I2T generation, we employ an autoregressive text decoder (Wang et al., 2022a), while for T2I generation we use a non-autoregressive parallel image decoder (Chang et al., 2023), which is an order of magnitude faster than autoregressive image decoders such as Yu et al. (2022b).

During training, ITIT utilizes a small set of paired image-text data to achieve reasonable text-to-image and image-to-text generation performance, and align concepts in the image and text spaces. Simultaneously, for unpaired image (text) data, ITIT generates corresponding text (image) counterparts and employs them as inputs to reconstruct the input image (text): this corresponds to a full cycle loss. We consider two kinds of full cycles: T2I2T (starting with an unpaired text sample); and I2T2I (starting with an unpaired image sample). For both modalities, we employ cross-entropy loss for the reconstruction, which enable a simple, unified approach for cycle consistency. ITIT enables us to leverage unpaired image and text data to provide informative supervision signals for training.

We evaluate the performance of ITIT on standard image-to-text and text-to-image generation benchmarks and demonstrate that, by leveraging unpaired data and cycle consistency, ITIT attains performance levels similar to a non-cycle baseline. However, ITIT uses significantly up to 2 orders of magnitude lower paired data. Furthermore, ITIT scales similarly with unpaired data as the baseline does with equivalent amounts of paired data, while being much more robust to low data quality. We also compare ITIT with state-of-the-art methods and show that we can achieve comparable performance on common text-to-image and image-to-text benchmarks with substantially lesser paired data. Our contributions are summarized as follows:

- We introduce a framework that unifies text-to-image and image-to-text generation, and propose ITIT, a novel technique that enforces consistency between cycle-generated images/text and their corresponding originals. This approach allows the training of image-to-text and text-to-image models using unpaired image and text data.

- We comprehensively evaluate the proposed ITIT framework and the image-text cycle consistency method, and demonstrate that they significantly enhance model performance.

- We show that ITIT can achieve performance on par with state-of-the-art methods on common text-to-image and image-to-text benchmarks with much lesser ($\sim$100x) paired data. When scaling up training data to improve model efficacy, we show that we can add only un-

paired examples using our framework and achieve similar performance as scaled-up paired data, without the downsides of significant manual effort and poor pairing quality.

## 2 LITERATURE REVIEW

**Image-to-Text Generation.** Various works explore autonomously generating textual descriptions from input images, either training the network with generative loss alone (Wang et al., 2022b; Alayrac et al., 2022; Chen et al., 2023; Li et al., 2022; 2023a), or combining it with contrastive learning (Yu et al., 2022a). GIT (Wang et al., 2022a) trains a model comprising an image encoder and an auto-regressive text decoder using a language modeling loss, the image encoder pre-trained with contrastive loss (Radford et al., 2021). In our work, we adopt a similar framework to GIT for our Image-to-Text (I2T) framework, but we initialize our image encoder from scratch.

**Text-to-Image Generation.** Recent works focus on two primary paradigms: diffusion-based models (Rombach et al. (2022); Dhariwal & Nichol (2021); Nichol et al. (2021); Saharia et al. (2022); Ramesh et al. (2022); Ruiz et al. (2023)); and token-based methods. Token-based strategies transform raw images into image tokens, and predict these tokens either in an autoregressive manner (Esser et al., 2021; Ramesh et al., 2021; Gafni et al., 2022; Yu et al., 2021; Ding et al., 2021; Yu et al., 2022b) or in parallel (Chang et al., 2022; Li et al., 2023b; Chang et al., 2023). Muse (Chang et al., 2023) demonstrates that token-based strategies with parallel decoding can be considerably faster than diffusion-based or autoregressive generative models. Since this speed advantage facilitates text-to-image synthesis during training, we adopt this strategy in our T2I framework.

**Unifying Image and Text Generation.** COBIT (You et al. (2023)) achieves this by employing distinct image and text unicoders, coupled with a unified cross-modal decoder. Additionally, CM3 (Aghajanyan et al. (2022)) and CM3Leon (Yu et al. (2023)) harness causally masked generative models trained on extensive multi-modal document datasets, and enable the synthesis of both text and images. However, all these works still heavily rely on large-scale *paired* image-text datasets.

**Leveraging Unpaired Data in Generative Vision-Language Training.** Early works have tried to use unpaired image and text to train image captioning model in an unsupervised way (Feng et al., 2019). However, the performance is relatively poor. Recent efforts in incorporating unpaired data into generative vision-language training primarily focus on pre-trained image and text encoders (Esser et al., 2021; Roberts et al., 2019). However, these applications are limited to pre-training and do not encompass the entire generative vision-language training procedure, thus providing only incremental improvements. In some cases, researchers have explored the use of text-only data to improve text decoders (Wang et al. (2022b)), utilizing text-to-text training. However, this only enhances the text decoder and not the image encoder, resulting again in constrained improvements.

**Cycle-consistency.** The concept of cycle consistency has previously been used to provide regularization and/or compensate for a lack of annotated data. Zach et al. (2010); Zhou et al. (2016); Godard et al. (2016); Zhu et al. (2017); Messikommer et al. (2022) explore it for computer vision applications such as learning dense correspondence, event detection, depth estimation, and image-to-image translation. Wang et al. (2023) and Goel et al. (2022) introduce regularization on image-text consistency during CLIP pre-training, aiming to learn a better visual representation. Most related to our work is Gorti & Ma (2018), which uses text-image-text cycle consistency to perform text-to-image translation, but the performance is poor. Moreover, none of the previous works has explored the potential of cycle consistency in generative vision-language training using unpaired data.

Our novel approach diverges from preceding vision-language models that heavily rely on either a large corpus of paired image-text data, or fine-tuning methods that target only text or image encoder/decoders separately. For the first time, our method facilitates the utilization of unpaired image and text data during generative vision-language training. This innovation significantly reduces the dependency on paired image-text samples during the training process, which empowers the expansion of generative vision-language training to nearly boundless text-only and image-only datasets.

## 3 METHOD

ITIT is the first framework that enables generative vision-language training on unpaired image-only and text-only data. It uses a simple yet effective architecture: a unified image-text encoder and two

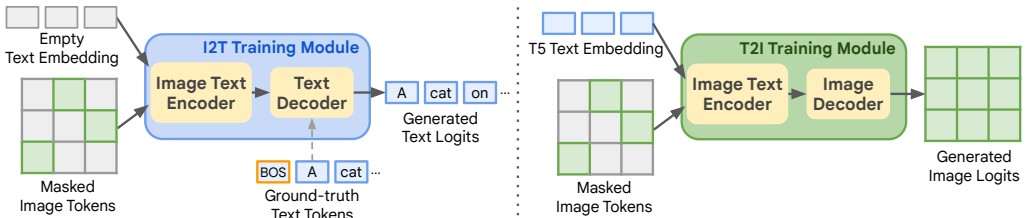

Figure 2: I2T (left) and T2I (right) training pipelines for *paired* image and text data.

separate image and text decoders. This design seamlessly enables text-to-image and image-to-text generation in the same framework, which paves the way for text-image-text (T2I2T) and image-text-image (I2T2I) cyclic losses. Below, we describe each component of our ITIT architecture and the cycle-consistency training paradigm in detail.

## 3.1 Unified Image-Text Generation Framework

**Architecture.** We first obtain text embedding $T = [t_l]_{l=1}^L$ from the output of a T5 encoder (Roberts et al., 2019) on the raw text. Similarly, raw images are passed through a pre-trained VQ-tokenizer (Esser et al., 2021) to output image tokens $I = [i_k]_{k=1}^K$. $L$ and $K$ are the token sequence lengths for text and image, respectively. The image tokens $I$ are then embedded with an embedding layer and concatenated with the T5 text features $T$ as input to the image-text encoder. Modality-specific decoders then operate on the encoded image-text features to generate either text or image tokens. The text decoder is autoregressive (Wang et al., 2022a), while the image decoder is parallel (Chang et al., 2023). Both encoder and decoders are based on Transformer (Vaswani et al., 2017) layers. A detailed description of the model architecture is included in Appendix B.

**Image-to-Text (I2T) Training.** As shown in Figure 2, we input masked image tokens along with empty text embedding to the image-text encoder. Masking is used to save computation, similar to MAE (He et al., 2022). We then use the features generated by the image-text encoder, as well as the ground-truth text tokens prepended with `[BOS]` (begin-of-sentence) token as the input to our text decoder. We use an auto-regressive language modeling (LM) loss to train the encoder and decoder:

$$\mathcal{L}_{I2T} = -\mathbb{E}_{(I,T)\in\mathcal{D}}\Big[\sum_{l=1}^L \log p(t_l|I_M, t_0, \cdots, t_{l-1})\Big], \tag{1}$$

which is a CE loss with label smoothing 0.1. Here, $t_0$ is set to be the `[BOS]` token. $I_M$ are the (subset of) *unmasked* tokens in $I$ and $p(i_k|I_M, T)$ is the probability predicted by the encoder-decoder network (the 'logits' layer), $\mathcal{D}$ is the distribution of paired image-text data. Note that the text decoder employs causal attention similar to GIT (Wang et al. (2022a)): each text token only depends on the preceding text tokens and all image features.

**Text-to-Image (T2I) Training.** As shown in Figure 2, right panel, we use masked image modeling for image generation, where the training objective is to reconstruct masked image tokens conditioned on the unmasked image tokens and the paired text features. We denote the binary mask determining which image tokens are masked by $M = [m_k]_{k=1}^K$. We use a cross-entropy loss between the ground-truth one-hot image tokens and the output of the image decoder. Specifically,

$$\mathcal{L}_{T2I} = -\mathbb{E}_{(I,T)\in\mathcal{D}}\Big[\sum_{\forall k: m_k=1} \log p(i_k|I_M, T)\Big], \tag{2}$$

**Inference.** We follow GIT (Wang et al., 2022a) for image-to-text inference and Muse (Chang et al., 2023) for text-to-image inference. More details are included in Appendix B.

## 3.2 Training with Cycle Consistency

Our cycle consistency training paradigm allows training with image-only and text-only data. The key idea is to first synthesize the corresponding text/image from the image-only or text-only data, and then use the synthesized data as input to reconstruct the original image/text. This allows us to apply cycle consistency supervision on image-only and text-only data.

**Text-Image-Text (T2I2T) Cycle.** Our T2I2T training pipeline is shown in Figure 3, top panel. At each training iteration, we first synthesize pseudo paired image tokens $I'$ for input text $T = [t_l]_{l=1}^L$

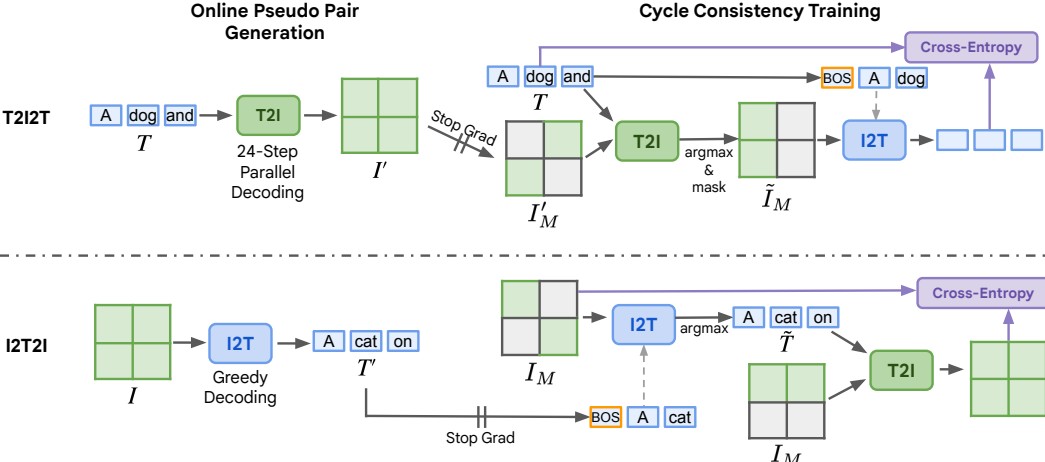

Figure 3: Text-image-text (top) and image-text-image (bottom) cycle training pipelines for *unpaired* image and text data. We use pseudo-generated image and text to enable the cycle consistency. Image token masks $M$ are always randomly chosen. The dashed line denotes causal attention. Text tokens prepended with [BOS] token are used for auto-regressive language modeling loss.

using our T2I inference pipeline. We then apply random mask $M$ to $I'$, perform reconstruction on $I'_M$ with the text $T$ using the T2I pipeline, and obtain the reconstructed synthesized image $\tilde{I}$. This two-step process allows us to avoid the excessive memory requirements of back-propagating gradients through all 24 steps of parallel decoding, while still training the T2I module. Finally, we randomly mask $\tilde{I}$ and use $\tilde{I}_M$ to generate text using the I2T pipeline. The objective of our cycle paradigm is to enforce consistency between this generated text and the original text. Therefore, the T2I2T cycle-consistency loss can be formulated as follows:

$$\mathcal{L}_{T2I2T} = -\mathbb{E}_{T \in \mathcal{D}_{text}}\Big[\sum_{l=1}^{L} \log p(t_l|\tilde{I}_M, t_0, \cdots, t_{l-1})\Big], \tag{3}$$

This is very similar to the I2T loss in Equation (1), except that $\tilde{I}$ is synthesized from $T$ instead of being drawn from the image-text joint distribution.

**Image-Text-Image (I2T2I) Consistency.** Our I2T2I training pipeline is shown in Figure 3, bottom panel. Similar to the T2I2T pipeline, we first synthesize pseudo paired text tokens $T'$ for input image tokens $I$ using our I2T inference pipeline. We then use the I2T training pipeline to predict $\hat{t}_l$ from $t'_0, \cdots, t'_{l-1}$ and $I_M$. As before, this avoids the excessive memory requirements of back-propagating gradients through the auto-regressive greedy decoding. We then mask $I$, and pass it through the T2I pipeline with the predicted $\tilde{T}$ to reconstruct the masked image tokens. Again, the loss enforces consistency between the reconstructed and the original image tokens using cross-entropy:

$$\mathcal{L}_{I2T2I} = -\mathbb{E}_{I \in \mathcal{D}_{image}}\Big[\sum_{\forall k: m_k=1} \log p(i_k|I_M, \tilde{T})\Big], \tag{4}$$

**Gradient Estimation.** One challenge in our cycle training is that $\tilde{i_k} = \arg\max(p(i_k|I'_M, T)$ and $\tilde{t_l} = \arg\max p(t_l|I_M, t'_0, \cdots, t'_{l-1})$, which are not differentiable. To solve this, we use a straight-through estimation on the predicted logits to approximate the gradient. Specifically, we directly copy the gradient on the one-hot prediction to the predicted logits after softmax. We show in section 4.4 that this helps improve both text-to-image and image-to-text performance.

# 4 RESULTS

## 4.1 EXPERIMENT SETUP

**Datasets.** We use three datasets in our experiments: CC3M (Sharma et al., 2018), WebLI (Chen et al., 2023), and Shutterstock (Shutterstock, 2023). CC3M contains 3.3 million high-quality image-text pairs. WebLI (Web Language Image) contains 111 million images where the image-text pairing

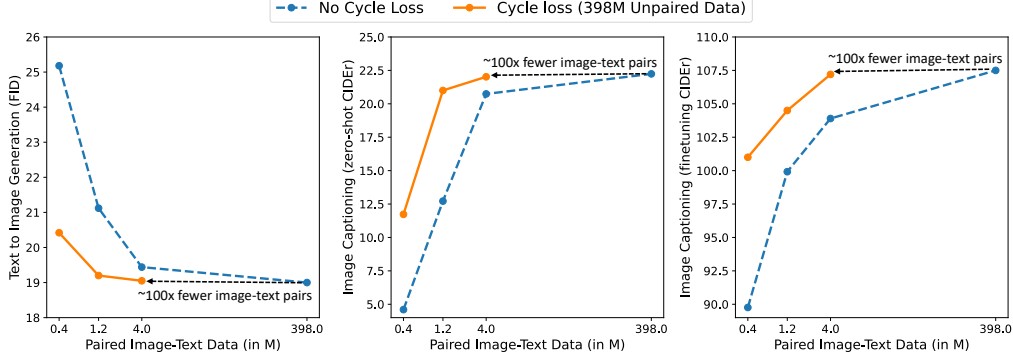

Figure 4: How ITIT-H's performance scales with additional paired Shutterstock data. The baseline (T2I+I2T) is trained with paired samples only. ITIT is trained with the same number of paired samples, as well as 398M unpaired samples (the full Shutterstock dataset) using cycle loss.

quality is much lower than CC3M. Thus, WebLI is significantly noisier and, as we show, leads to worse performance for I2T. Shutterstock contains 398 million images labeled by human annotators, which incurs significant expense and effort. More dataset details are included in Appendix C.

We use CC3M as our paired dataset, 50% of WebLI images as our unpaired image dataset, and the other 50% of WebLI texts as our unpaired text dataset for most of our experiments (Section 4.3 and Section 4.4). This 50%-50% split ensures that corresponding image-text pairs are not present in our unpaired image and text splits. We use the Shutterstock dataset in Section 4.2, where we analyze how ITIT scales w.r.t. different number of paired and unpaired data samples.

**Training.** We set the input image resolution as 256x256 to be consistent with previous literature. After passing through the VQGAN tokenizer, the image token sequence length is 16x16 (256 tokens). The raw text (maximum length of 64) is tokenized by SentencePiece tokenization (SentencePiece, 2023), and embedded using a pre-trained T5 encoder. These embeddings are then concatenated with the image token embeddings as the input to our image-text encoder.

We experiment with ViT-B, ViT-L, and ViT-H size Transformers (Dosovitskiy et al. (2021)) for our image-text encoder. We combine the losses in Equations 1 through 4 with equal weight for training. For results in Section 4.3, we use Adafactor (Shazeer & Stern, 2018) to train the model for 1.5M steps with a batch size of 2048 (1024 for image-text pairs, 512 for unpaired images, and 512 for unpaired texts). We use a cosine learning rate schedule with 5K steps warmup and maximum learning rate $1 \times 10^{-4}$. For other experiments, we use the exact same training paradigm except that we train the models for 500K steps. More details are included in Appendix B.

**Evaluation.** We follow the commonly used MS-COCO benchmark and evaluation protocols. For image-captioning, we evaluate both the zero-shot and fine-tuning performance of ITIT on the COCO Karpathy split (Karpathy & Fei-Fei, 2015) and report the CIDEr score (Vedantam et al., 2015). For text-to-image generation, we evaluate ITIT on 30K image-text pairs randomly selected from the COCO Captions training set and report the Frechet Inception Distance (FID) score (Heusel et al., 2017). CIDEr is the higher the better, and FID is the lower the better.

## 4.2 SCALE WITH DATA

In this section, we comprehensively evaluate ITIT's performance with different amounts of paired and unpaired data on Shutterstock dataset (Shutterstock, 2023) consisting of 398M image-text pairs.

Figure 4 analyses how ITIT's performance scales with paired data. We train a baseline with only paired data, with the sum of the losses in Equation (1) and Equation (2). ITIT is trained with the same paired data as the baseline, and the entire set of 398M images and text present in Shutterstock as unpaired data. More paired data helps both settings, but training with unpaired data significantly improves ITIT's performance over the baseline on both image captioning and text-to-image generation. Remarkably, with only 4M paired data and 398M unpaired data, ITIT achieves *a similar performance as training with 398M paired data*. Note that ITIT does not use any samples not present in the baseline trained with 398M paired data, as all of the samples are from Shutterstock. Therefore

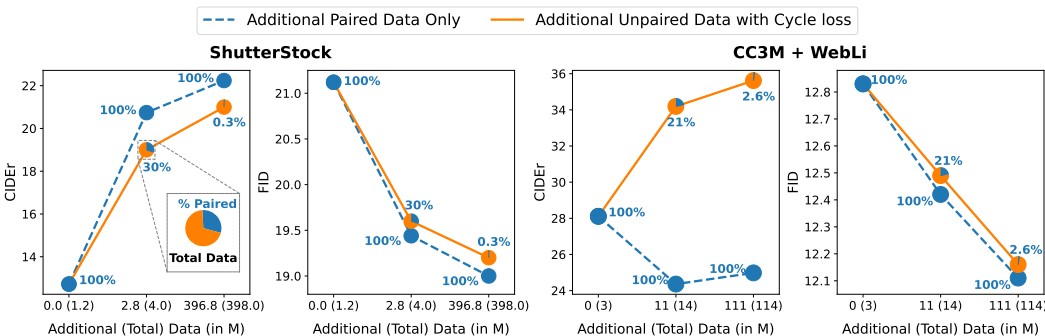

Figure 5: How ITIT's performance scales with the total amount of data used (x-axis). The base-line (T2I + I2T) in blue is trained entirely with increasing amounts of paired data. ITIT (orange) is trained with an increasing amount of unpaired data using cycle loss, while keeping the total amount of data equal for both curves. For example, the rightmost point with Shutterstock uses 1.2M image-text pairs and 396.8M unpaired samples (half as unpaired image and half as unpaired text) for ITIT with cycle loss, and 398M image-text pairs for the baseline. *Left*: Shutterstock data as both paired and unpaired. *Right*: CC3M as paired data, and varying fractions of WebLI as additional paired / unpaired data.

ITIT can perform similarly as a baseline with 100x fewer image-text pairs, significantly reducing the effort and expense for the training of generative vision-language training.

Next, we evaluate how ITIT's performance scales w.r.t. the total amount of data used. We first train a model with 1.2M paired image-text Shutterstock data. We then evaluate the effect of training models on adding increasing amounts of additional paired data vs. adding increasing amounts of unpaired data with cycle loss, keeping the total amount of data the same for both. As expected, we see in Figure 5 that performance scales up with additional paired data. Surprisingly, however, additional unpaired data exhibits similar scalability. In fact, we can achieve 19.2 FID and 21.0 CIDEr with only 1.2M paired and 396.8M unpaired examples, which is very competitive with 19.0 FID and 22.2 CIDEr using 398M paired examples only. This experiment thus demonstrates that when scaling up training data, practitioners can rely on only adding unpaired examples using our method and achieve similar performance as paired data without the extra manual effort required to collect it.

We repeat the above experiment in a more realistic setting, where the small-scale paired dataset can contain high-quality image-text pairs but a large-scale paired dataset has much lower quality. For this, we use the high-quality CC3M as the paired dataset, and the much larger WebLI as the low-quality unpaired dataset. As before, we start with a model trained on 3M paired examples (from CC3M), and add additional training data from WebLI in paired (blue) or unpaired (orange) form. As shown in Figure 5, right pair, adding low-quality image-text pairs harms image captioning performance severely for the fully-paired case. However, the ITIT regime is not affected by this low quality and scales similarly as before. This demonstrates that our method is robust to low data quality in large datasets, and can in fact be used to achieve significantly better performance in settings when paired data is present but of low quality.

### 4.3 COMPARISON TO PRIOR WORK

In Table 1, we compare ITIT with state-of-the-art image-to-text and text-to-image models on the commonly used MS-COCO benchmark. As shown, all SOTA methods rely heavily on training on a large corpus of paired image-text data. ITIT, however, is trained with only 3M paired examples (CC3M), and an additional 55M unpaired image and text examples each (WebLI). Despite this, it beats many other methods trained on much more data for text-to-image generation (FID). For I2T, it beats methods using a comparable amount of data (highlighted in green), and achieves performance competitive with other SOTA methods. We find that the pre-training data (both the mixture and the size) also makes a difference to CIDEr score. For example, GIT (Wang et al., 2022a) achieves only 89.0 CIDEr fine-tuning performance on COCO captions when trained from scratch with 10M image-text pairs, which is far from its reported performance (144.8) when trained with 800M image-text pairs. Our approach is orthogonal to dataset mixture considerations, and we believe that scaling data size and variety will further enhance FID and CIDEr scores. We leave this to future work.

Table 1: Quantitative comparison with state-of-the-art text-to-image and image-to-text models on MS-COCO. The image-captioning performance is evaluated on the COCO Karpathy split, and the text-to-image generation FID is evaluated on 30K COCO images. † denotes our re-implementation. We highlight in green other models that use comparable amounts of paired data. Note that the GIT (CLIP) model uses a CLIP (Radford et al., 2021) encoder pre-trained with 400M image-text pairs.

| Methods | #params | #paired data | #unpaired data | FID↓ | CIDEr↑ (zs) | CIDEr↑ (ft) |
|---|---|---|---|---|---|---|
| *T2I* | | | | | | |
| StableDiffusion (Rombach et al., 2022) | 800M | 400M | - | 12.60 | - | - |
| GLIDE (Nichol et al., 2021) | 5B | 250M | - | 12.24 | - | - |
| Make-A-Scene (Gafni et al., 2022) | 4B | 35M | - | 11.84 | - | - |
| DALL-E 2 (Ramesh et al., 2022) | 3.5B | 650M | - | 10.39 | - | - |
| PARTI (Yu et al., 2022b) | 750M | 5000M | - | 10.71 | - | - |
| Muse-512 (Chang et al., 2023) | 3B | 860M | - | 7.88 | - | - |
| Muse† (Chang et al., 2023) | 750M | 3M | - | 23.7 | - | - |
| *I2T* | | | | | | |
| BLIP (Li et al., 2022) | 446M | 129M | - | - | - | 136.7 |
| SimVLM$_{base}$ (Wang et al., 2022b) | - | 1100M | 365M T | - | 24.0 | 134.8 |
| SimVLM$_{huge}$ (Wang et al., 2022b) | ~1.4B | 1100M | 365M T | - | 32.2 | 143.3 |
| GIT (CLIP) (Wang et al., 2022a) | 681M | 800M | - | - | - | 144.8 |
| GIT$_B$ (scratch) (Wang et al., 2022a) | 129M | 10M | - | - | - | 89.0 |
| *T2I+I2T* | | | | | | |
| CoBIT-Base (You et al., 2023) | 626M | 5200M | - | 10.35 | 43.0 | 135.4 |
| CoBIT-Large (You et al., 2023) | 1091M | 5200M | - | 9.37 | 44.8 | 139.5 |
| CM3Leon (Yu et al., 2023) | 7B | 340M | - | 4.88 | 61.6 | - |
| ITIT-B | 221M | 3M | 55M I+55M T | 13.4 | 32.1 | 103.5 |
| ITIT-L | 487M | 3M | 55M I+55M T | 12.0 | 35.1 | 116.4 |
| ITIT-H | 868M | 3M | 55M I+55M T | 10.4 | 38.2 | 125.3 |

## 4.4 Ablations

In Table 2, we ablate the effectiveness of the four components of ITIT: T2I, I2T, T2I2T, and I2T2I. As shown in rows 1-3, combining T2I and I2T training in our framework already improves image captioning performance. This is likely because the T2I training alleviates the overfitting problem of I2T training, as shown in GIT (Wang et al., 2022a).

As before (Figure 5), we can see in row 4 that combining CC3M and WebLI improves text-to-image generation, but harms image captioning performance. This is because of the lower image-text pairing quality of WebLI compared to CC3M. The remaining rows demonstrate that the cycle loss alleviates this by using WebLI as unpaired data and does not depend on its image-text pairing quality. It is thus more generalizable to large-scale image-text datasets.

Next, rows 5-7 are naive baselines for using unpaired image or text data during generative vision-language training. We can simply perform text-to-text (T2T) autoregressive training without conditioning on images, which has been explored in some prior works (Wang et al. (2022b)). Similarly, we can perform image-to-image (I2I) reconstructive training without conditioning on text. Such baselines do improve the performance over not using any paired data (row 3).

We consider an ablation where the gradient of the cycle consistency loss is backpropagated up until the argmax step. Hence, only half of the cycle is trained. In fact, this is equivalent to first synthesizing an image counterpart from unpaired text and then using it as a pseudo image-text pair to train the I2T model (similarly for T2I). Rows 8-10 show that the half-cycle loss achieves much better performance than non-cycle baselines.

Finally, rows 11-14 show the performance of the full cycle ITIT training. Although T2I2T favors image captioning while I2T2I favors text-to-image generation, they both show significant improvement in text-to-image generation and image captioning. Moreover, row 14 demonstrates that such two cycle losses can be combined to further improve performance. Additionally, we can see that the full cycle loss beats the half-cycle baselines (row 8-10), demonstrating the effectiveness of the gradient estimation step.

Lastly, we find by comparing row 3 and 13 that the cycle consistency loss can slightly improve the performance even without any additional data. We believe this is because it forces better image-text alignment. However, comparing row 13 and 14 shows that the huge improvements in both text-to-image and image-to-text generation mainly stem from the usage of additional unpaired data.

Table 2: Quantitative comparison between different variants of ITIT on MS-COCO. All experiments use ITIT$_B$ trained with 500K steps. We take 50% of WebLI data and use the images as our unpaired image data, and the other 50% of WebLI data and use the texts as our unpaired text data.

| | T2I | I2T | T2I2T | I2T2I | paired data | unpaired text | unpaired image | FID↓ | CIDEr↑ |
|---|---|---|---|---|---|---|---|---|---|
| *Paired data only* | | | | | | | | | |
| 1 | ✓ | ✗ | ✗ | ✗ | CC3M | ✗ | ✗ | 15.5 | N/A |
| 2 | ✗ | ✓ | ✗ | ✗ | CC3M | ✗ | ✗ | N/A | 19.0 |
| 3 | ✓ | ✓ | ✗ | ✗ | CC3M | ✗ | ✗ | 15.7 | 23.5 |
| 4 | ✓ | ✓ | ✗ | ✗ | CC3M+WebLI | ✗ | ✗ | 14.2 | 20.7 |
| *Paired+unpaired data, no cycle* | | | | | | | | | |
| 5 | ✓ | ✓ | T2T | ✗ | CC3M | 50% WebLI | ✗ | 15.1 | 26.0 |
| 6 | ✓ | ✓ | ✗ | I2I | CC3M | ✗ | 50% WebLI | 15.9 | 24.2 |
| 7 | ✓ | ✓ | T2T | I2I | CC3M | 50% WebLI | 50% WebLI | 15.6 | 28.5 |
| *Paired+unpaired data, half cycle* | | | | | | | | | |
| 8 | ✓ | ✓ | Half | ✗ | CC3M | 50% WebLI | ✗ | 14.8 | 27.6 |
| 9 | ✓ | ✓ | ✗ | Half | CC3M | ✗ | 50% WebLI | 14.7 | 24.8 |
| 10 | ✓ | ✓ | Half | Half | CC3M | 50% WebLI | 50% WebLI | 14.5 | 30.5 |
| *Paired+unpaired data, full cycle* | | | | | | | | | |
| 11 | ✓ | ✓ | Full | ✗ | CC3M | 50% WebLI | ✗ | 14.6 | 28.4 |
| 12 | ✓ | ✓ | ✗ | Full | CC3M | ✗ | 50% WebLI | 14.6 | 26.3 |
| 13 | ✓ | ✓ | Full | Full | CC3M | CC3M | CC3M | 15.4 | 24.4 |
| 14 | ✓ | ✓ | Full | Full | CC3M | 50% WebLI | 50% WebLI | **14.3** | **31.1** |

Figure 6: Iteratively generating text to image to text and so on. With ITIT, the generated results are more consistent than the results from a model trained without the cycle consistency loss.

## 4.5 CYCLE-GENERATION RESULTS

With a framework that can perform both image-to-text and text-to-image, we can easily perform cycle-generation, as shown in Figure 6. With ITIT training, the cycle generation often keeps the same semantics as the input text prompt. On the other hand, without the cycle consistency training, the cycle generation misses the "blue" semantics after the first cycle. This demonstrates that our cycle consistency training not only enables integrating unpaired image and text data into generative vision-language training, but also improves image-text alignment for both image-to-text and text-to-image generation. We include a number of results of image and text generation in Appendix A (Figures 1 through 4).

## 5 DISCUSSION

We propose ITIT, a novel training scheme that for the first time incorporates unpaired images and text into generative vision-language training. Through extensive ablations, we demonstrate the effectiveness of both the T2I2T cycle and I2T2I cycle in improving text-to-image and image-to-text generation performance. As a result, ITIT achieves performance competitive with state-of-the-art vision-language generative models, but with only 3 million paired image-text samples. Our method can be used even when paired image-text data is present, and is especially helpful when the pairing quality is low. Future directions include scaling ITIT to larger unpaired image and text data and model sizes, and utilizing more diverse datasets.

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

# 6 ADDITIONAL QUALITATIVE RESULTS

**Image-to-Text and Text-to-Image Generation.** In Figure 7, Figure 8 and Figure 9, we show the performance of ITIT on text-to-image generation and image captioning. The model uses ViT-H as the backbone and is trained with CC3M as paired data and WebLI as unpaired data for 1.5M steps. As shown in the results, our model can generate realistic images from text prompts, and can also generate accurate captions for the input image.

**Cycle Generation.** We include more cycle-generation results in Figure 10. With ITIT, the generated results are quite consistent with the original input. Without the cycle consistency loss, the generated text/image can easily miss some key information in the input image/text, causing the cycle-generation results to diverge from the original input. This demonstrates that the proposed cycle consistency loss forces the cycle generation to be more consistent and improve the input-output alignment for both text-to-image and image-to-text generation.

# 7 IMPLEMENTATION DETAILS

In this section, we include our implementation details, including hyper-parameters, model architecture, and training paradigm.

**Image Tokenizer and Detokenizer.** We use a CNN-based VQGAN encoder to encode the 256x256 input images to 16x16 feature maps. The quantizer then quantizes each pixel of the encoder's output feature map using a codebook with 8192 entries. The detokenizer operates on the 16x16 discrete tokens and reconstructs the 256x256 image. Our VQGAN tokenizer and detokenizer are trained on the WebLI dataset with batch size 256.

**ViT architecture.** After the tokenizer, the image latent sequence length becomes 256. Since we always use a masking ratio larger than 50%, we drop 128 of the masked image tokens from the input. The tokens are then embedded with an embedding layer and concatenated with the text embedding from T5-XXL with a length of 64. We then use an image-text encoder Transformer with standard ViT architecture Dosovitskiy et al. (2021), which consists of a stack of Transformer blocks Vaswani et al. (2017), where each block consists of a multi-head self-attention block and an MLP block.

The text decoder is similar to the one used in GIT (Wang et al., 2022a), which consists of 6 Transformer blocks with causal attention mask. The attention mask ensures each text token can only attend to its previous text tokens and the image tokens.

The image decoder is similar to the one used in MAGE (Li et al., 2023b), which consists of 8 Transformer blocks with self-attention. The input to the image decoder is padded with the previously dropped masked image tokens. In this way, we save a lot of computation in the image-text encoder.

**Vision-language Training.** Please refer to Table 3 for our default vision-language training setting. With the online synthesis step, ITIT requires $\sim$2x training time as standard I2T and T2I non-cycle training. Our ViT-H training with 1.5M steps takes $\sim$10.9 days on 512 TPUv3.

**Gradient scale for I2T loss:** Similar to GIT (Wang et al. (2022a)), we found that the image-text encoder should receive a smaller gradient than the text decoder. Therefore, we scale down the gradient backpropagated from the text decoder to the image-text encoder by 0.1. In practice, this is simply implemented by $z = 0.1z + \text{stopgrad}(0.9z)$ where $z$ denotes the output of the image-text encoder.

**Cycle Training.** During cycle training, the image-text encoder experiences the forward pass twice. We found that if we back-propagated the gradient back to it twice, the training becomes unstable. Therefore, we stop the gradient after the first forward pass of the image-text encoder, which significantly stabilize the cycle training.

**Gradient Estimation.** We use Gumbel softmax (Jang et al., 2016) with strength 1.0 for the gradient estimation in the T2I2T cycle, and use straight-through softmax for the gradient estimation in the I2T2I cycle. We found that using Gumbel softmax in the T2I2T cycle improves zero-shot CIDEr by 0.3, while using Gumbel softmax in the I2T2I cycle does not bring improvement.

**I2T Inference.** To generate captions for an image, we first extract visual features from the image tokens (and empty text tokens) using the image-text encoder. Then we auto-regressively generate the

Table 3: **Pre-training Setting.**

| config | value |
| --- | --- |
| optimizer | Adafactor (Shazeer & Stern, 2018) |
| peak learning rate | 1e-4 |
| weight decay | 0.045 |
| optimizer momentum | $\beta_1, \beta_2 = 0.9, 0.96$ |
| T2I batch size | 512 |
| I2T/I2I batch size | 512 |
| T2I2T batch size | 512 |
| I2T2I batch size | 512 |
| learning rate schedule | cosine decay (Loshchilov & Hutter, 2016) |
| warmup steps | 5000 |
| training steps | 1.5M |
| gradient clip | 3.0 |
| label smoothing (Szegedy et al., 2016) | 0.1 |
| dropout | 0.1 |
| image masking ratio min | 0.5 |
| image masking ratio max | 1.0 (T2I), 0.75 (I2T) |
| image masking ratio mode | 0.75 |
| image masking ratio std | 0.25 |

tokens, conditioned on the visual features and previously generated tokens. Following GIT (Wang et al. (2022a)), we use beam search to generate the tokens with a beam size set to 4.

**T2I Inference.** To enable classifier-free guidance (Ho & Salimans, 2022) for generation, when performing I2T training with empty text embedding as input, we also train the image decoder to reconstruct missing tokens. In such cases, the T2I training becomes image-to-image (I2I) training, which is to reconstruct the original tokens from unmasked tokens.

Similar to Muse (Chang et al. (2023)), we use parallel decoding with classifier-free guidance to generate an image from a text prompt. We start with entirely masked image tokens, and concatenate them with the text prompt tokens. At each iteration, the decoder predicts a conditional logit $l_c$ and an unconditional logit $l_u$ for each masked token. The final logits $l_g$ are formed by moving away from $l_u$ by the guidance scale $\tau$: $l_g = (1 + \tau)l_c - \tau l_u$. We then sample each token using categorical sampling. After that, the corresponding prediction score of each token plus a noise sampled from a random Gumbel distribution multiplied by temperature $t$ is used as the "confidence" score indicating the model's belief of each token prediction. We then sample the top-k tokens with the highest predicted probability, and replace the corresponding masked tokens with these sampled predicted tokens. The number of masked tokens to be replaced in each iteration follows a cosine function (Chang et al., 2022). We use a total of 24 steps to generate an image. For each model, we sweep temperature $t$ and classifier-free guidance scale $\tau$ for the optimal FID. In practice, we find $t = 32$ and $\tau = 2.0$ serve as near-optimal temperature and guidance scale.

# 8 DATASETS

We use three datasets in our main paper: CC3M, WebLI, and Shutterstock. CC3M contains 3.3M image-text pairs. The raw descriptions are harvested from the alt-text HTML attribute associated with web images and then filtered to form the final dataset. WebLI (Web Language Image) contains 111 million images from the public web with image-associated alt-text labels where the image-text pairing quality is much lower than CC3M, because no filtering was applied to the alt-text data. Shutterstock contains 398 million images labeled by human annotators, which incurs significant expense and effort.

# 9 TRAINING SPEED

In Table 4, we show the training speed of different variants of ITIT. All experiments are evaluated on a cluster of 256 TPUv4, with total batch size equals 2048. As shown in the table, ITIT is ~2.3× slower than the non-cycle baseline. The full cycle ITIT is slightly slower than the half cycle ITIT as the gradient needs to back-propagate before the argmax. Despite doubling the training time, ITIT significantly reduce the need of collecting large-scale paired datasets. We show in our paper

Table 4: Training speed of different variants of ITIT in steps/s. All experiments use $ITIT_B$ with total batch size of 2048 and are evaluated using a cluster of 256 TPUv4.

| | T2I | I2T | T2I2T | I2T2I | Steps/s |
|---|---|---|---|---|---|
| *No cycle* | | | | | |
| 1 | ✓ | ✓ | ✗ | ✗ | 3.41 |
| *Half cycle* | | | | | |
| 2 | ✓ | ✓ | Half | ✗ | 2.41 |
| 3 | ✓ | ✓ | ✗ | Half | 2.19 |
| 4 | ✓ | ✓ | Half | Half | 1.64 |
| *Full cycle* | | | | | |
| 5 | ✓ | ✓ | Full | ✗ | 2.25 |
| 6 | ✓ | ✓ | ✗ | Full | 2.05 |
| 7 | ✓ | ✓ | Full | Full | 1.47 |

that ITIT trained with only 4M paired data can achieve a similar performance as baseline trained with 400M paired data, reducing the cost of collecting paired data by 100 times.

## 10 WARM-UP

Table 5: Warm-up steps. Incorporating cycle consistency loss at 25K steps achieves the best performance.

| Warm-up steps | 0 | 25K | 50K | 100K | 200K |
|---|---|---|---|---|---|
| FID | 14.3 | **14.1** | 14.3 | 14.6 | 14.8 |
| CIDEr | 31.1 | **31.3** | 31.3 | 30.9 | 30.5 |

At the beginning of the training process, the text-to-image (T2I) and image-to-text (I2T) modules struggle to generate realistic images or texts. Introducing cycle consistency loss during this initial phase would likely introduce unnecessary noise into the training process. To address this, we experimented with a warm-up training scheme, as shown in Table 5. This approach involves delaying the introduction of cycle consistency loss until the T2I and I2T modules have undergone several steps of training. We evaluated all experiments using an ITIT-B model trained for 500K steps. The results indicate that incorporating cycle consistency loss after 25K steps yields optimal performance. Interestingly, introducing the cycle loss at the very beginning (0 step) does not significantly impair performance. This resilience is likely due to the dominance of paired data in the early training stages. As the generation results improve, the impact of cycle consistency loss becomes more pronounced, achieving greater diversity and generalization ability through the utilization of unpaired data.

## 11 FAILURE CASES

In Figure 11, we present various instances where the ITIT model does not perform as expected. The first notable type of failure involves the generation of text within images. Additionally, the model occasionally produces images with watermarks or margins, artifacts that can be traced back to the training data. Another challenge arises when the model attempts to generate multiple objects within a single image, often resulting in a compromise on the generation quality of each individual object.

It is important to recognize, however, that these shortcomings are not unique to ITIT; they are, in fact, common across most image-to-text generation models. Some of these issues, such as watermarks, are closely tied to the quality of the data utilized in training. Given this context, we believe that ITIT has the potential to mitigate these prevalent shortcomings by lessening the dependency

on paired data in vision-language generative training. A promising direction for future research would involve training ITIT with a combination of a small but exceptionally high-quality, paired image-text dataset and extensive, high-quality unpaired image and text datasets. This approach could significantly enhance the model's performance by providing a richer and more diverse training environment, potentially overcoming the common failure cases that currently hinder image-to-text generation models.

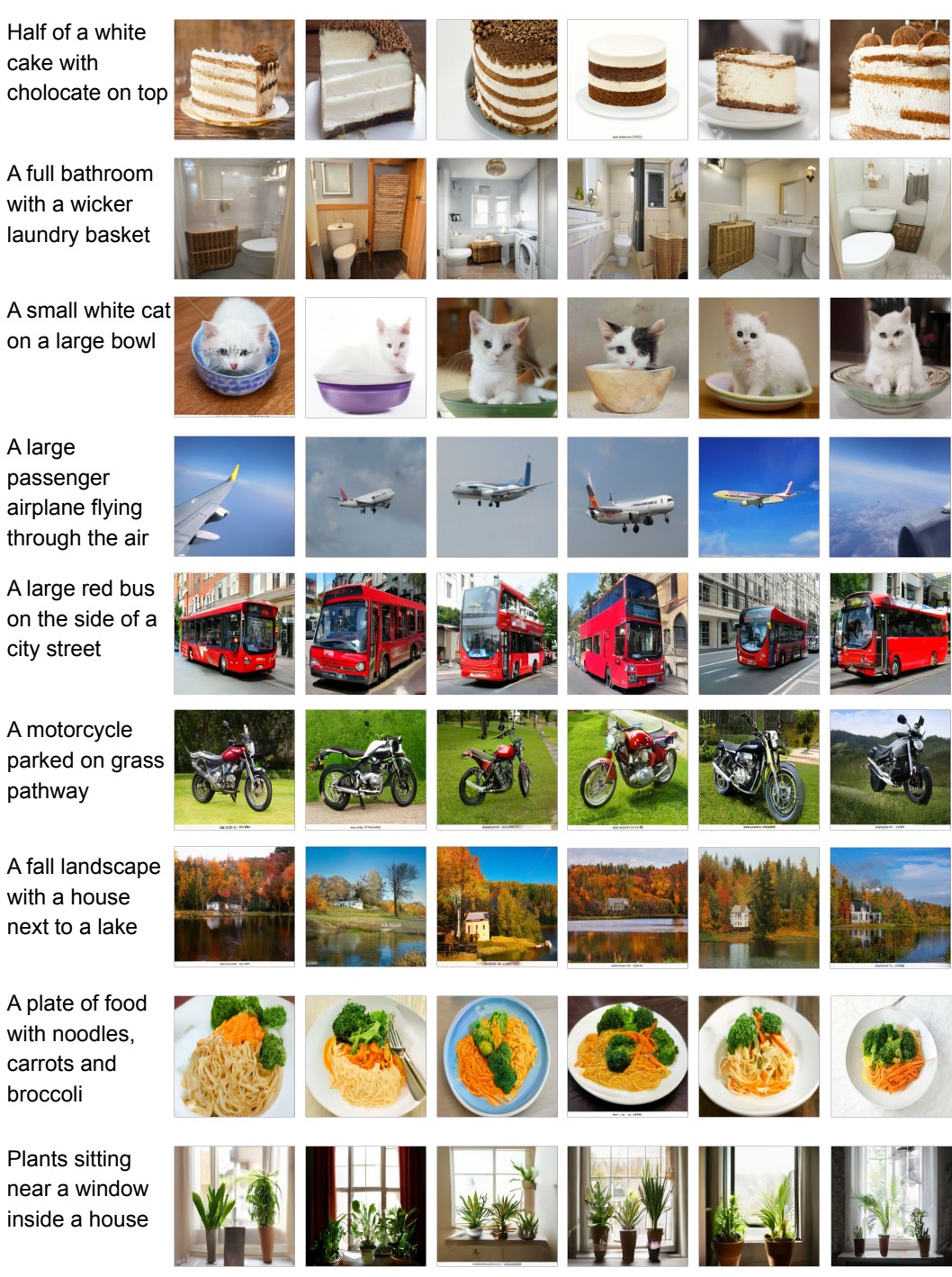

Figure 7: ITIT text-to-image generation results.

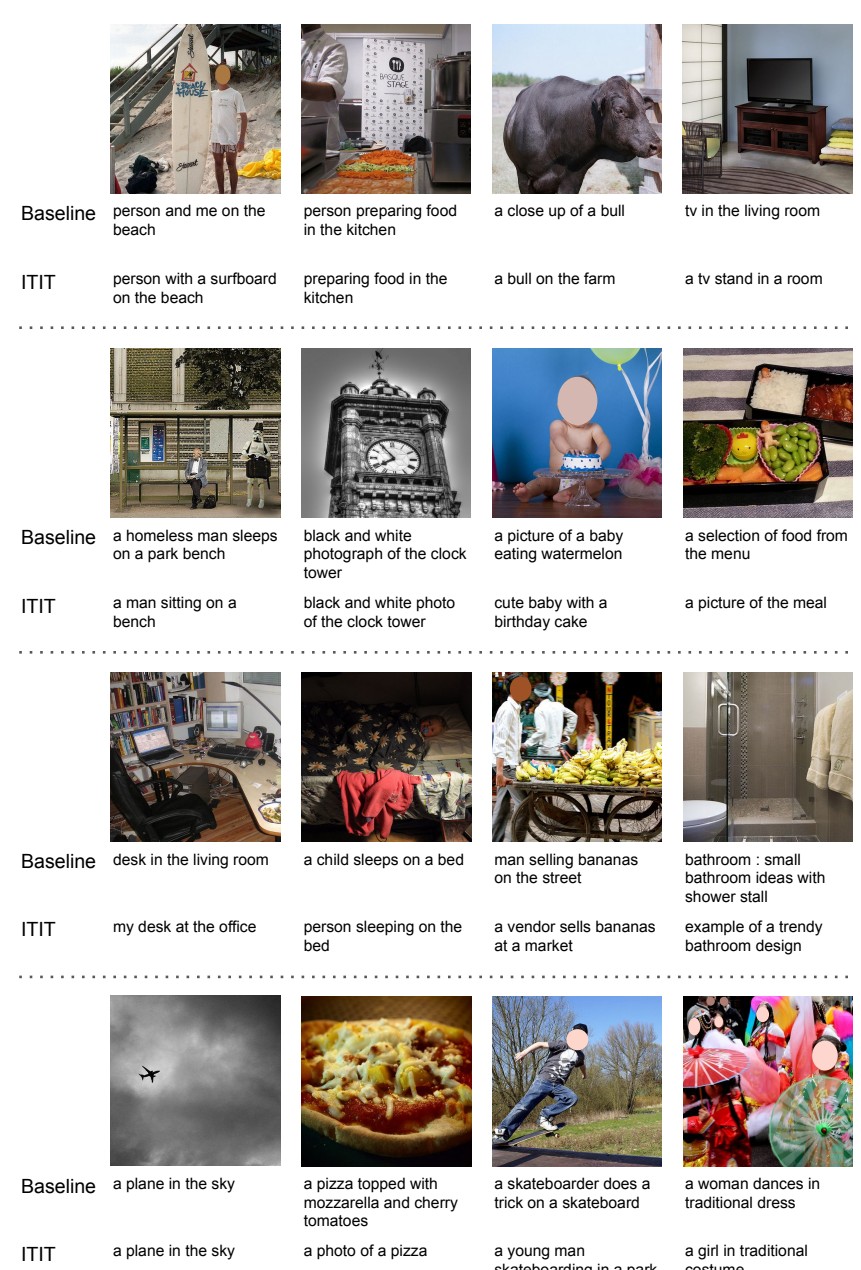

Figure 8: Image-to-text generation performance (zero-shot on COCO Captions). The baseline is trained on CC3M only. ITIT is trained on CC3M as paired data and WebLI as unpaired data. We obfuscate the faces of people in the images.

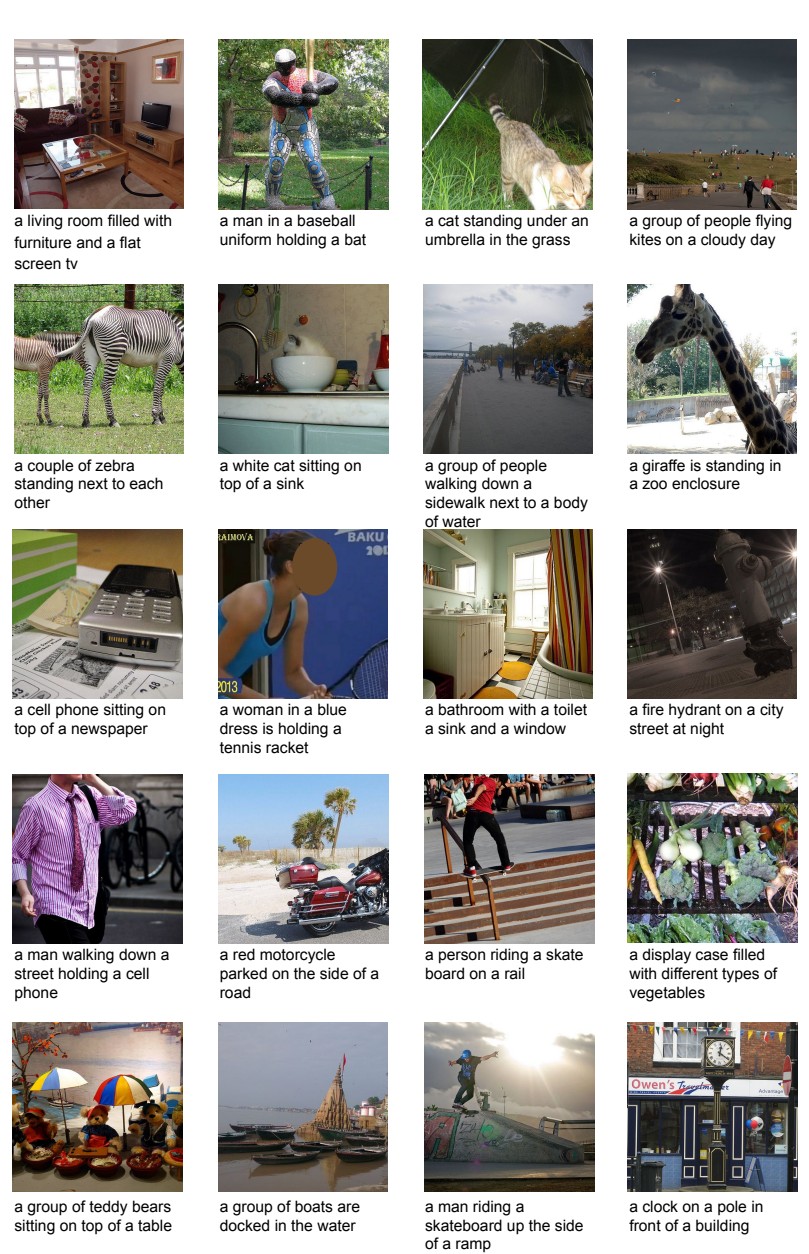

Figure 9: ITIT image-to-text generation results (fine-tuned on COCO Captions).

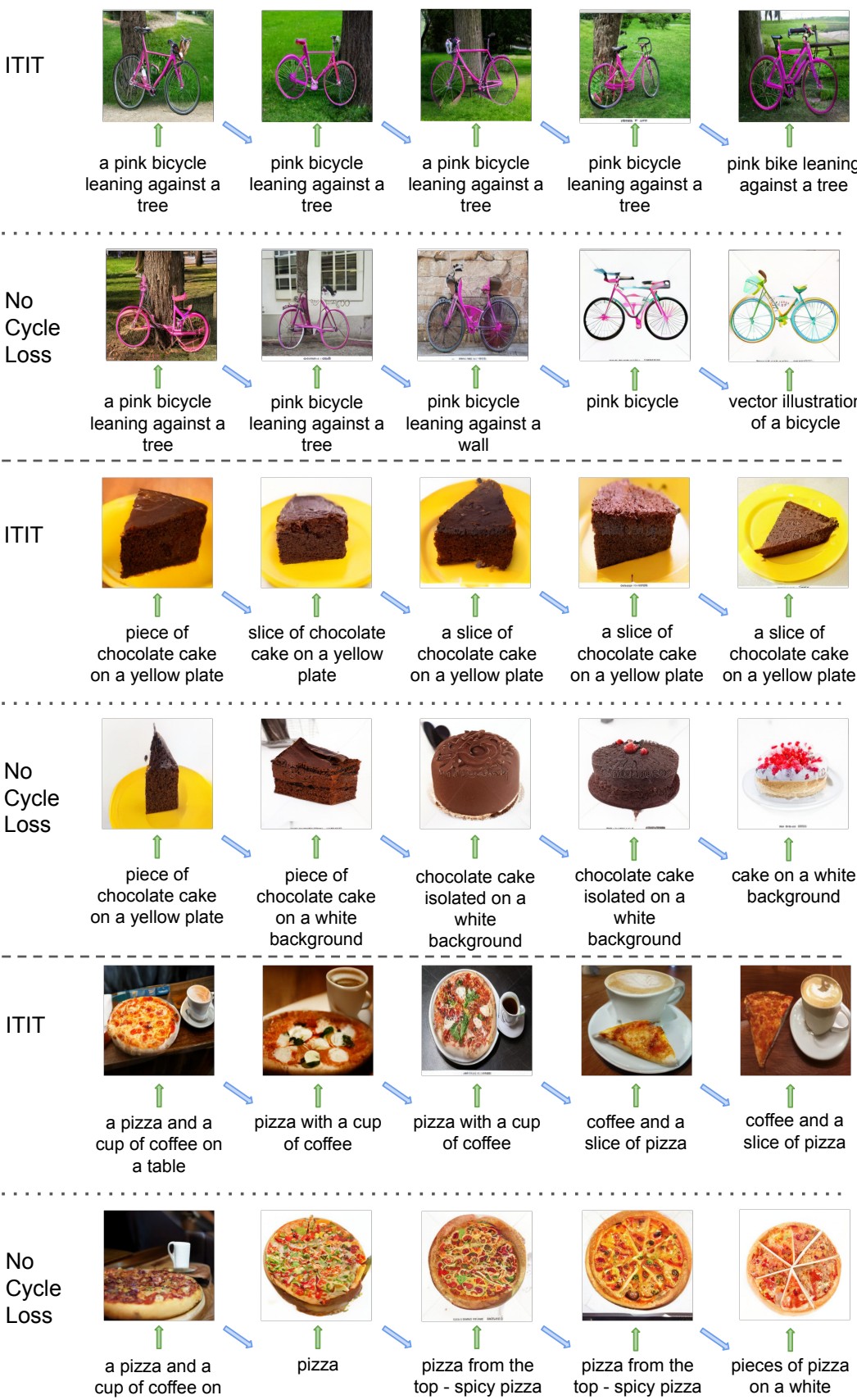

Figure 10: More cycle generation results with or without cycle consistency loss.

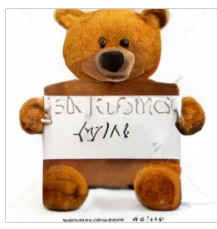 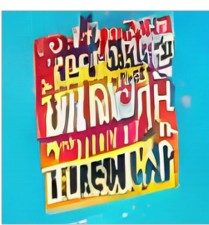 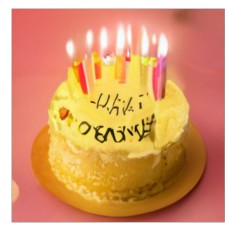 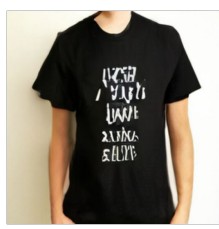

| A bear holding a board with 'hello' written on it | A colorful slogan | A cake with 'happy birthday' on it | A t-shirt with slogan on it |

(a) Generate text in image

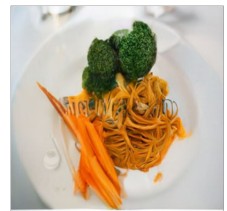 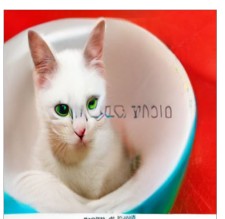 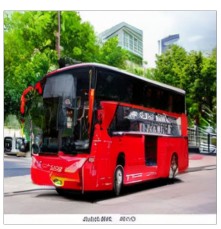 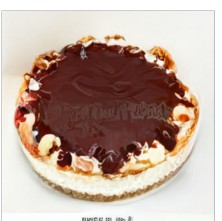

| A plate of food with noodles, carrots and broccoli | A small white cat on a large bowl | A large red bus on the side of a city street | White cake with cholocate on top |

(b) Watermarks / margins in the dataset

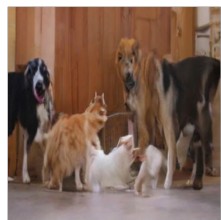 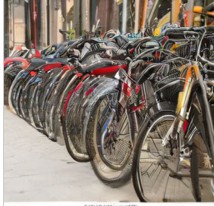 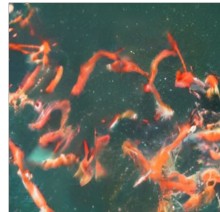 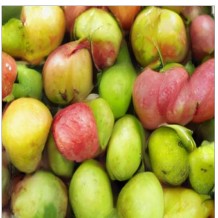

| Five cats and five dogs are playing together | Many bicycles on the street | Many fishes in a pool | Many apples and pears |

(c) Multiple objects in one image

Figure 11: ITIT failure cases.

