# OpenReview forum: "Leveraging Unpaired Data for Vision-Language Generative Models via Cycle Consistency"
_ICLR.cc/2024/Conference — ICLR 2024 spotlight_

### Official Review · Reviewer_NW6v · 2023-10-22

**Soundness:** 4 excellent
**Presentation:** 3 good
**Contribution:** 4 excellent
**Rating:** 8
**Confidence:** 3

**Summary:**

The study introduces a groundbreaking training paradigm called ITIT, designed for vision-language generative models. ITIT capitalizes on cycle consistency to train on unpaired image and text datasets, thereby minimizing reliance on costly and extensive paired datasets. The architecture features a single encoder for both image and text data, along with distinct decoders for each. A minimal amount of paired data is employed for baseline training, while the bulk of the training leverages unpaired data, regulated by cycle consistency to maintain output fidelity. Experimental evaluations reveal that ITIT achieves performance levels similar to existing leading models but with a significantly reduced need for paired data. Furthermore, the study confirms the model's ability to effectively scale with unpaired data. ITIT paves the way for more efficient and economical training of vision-language models.

**Strengths:**

1. The introduction of cycle consistency in vision-language generative models is innovative and addresses a significant gap in the literature. This is particularly relevant for scenarios where paired data is scarce or expensive to obtain.

2. ITIT scales well with unpaired data, achieving performance levels similar to non-cycle baselines but with much lower paired data requirements. In addition, the paper provides a thorough evaluation of the proposed ITIT framework, demonstrating its efficacy in both image-to-text and text-to-image generation tasks with comprehensive experiments.

3. The cycle consistency training improves image-text alignment for both image-to-text and text-to-image generation, while the method without the cycle consistency loss fail on the generation consistency to some extent.

**Weaknesses:**

1. While the paper mentions techniques to reduce computational overhead by one-step back-propagation in the I2T2I cycle, it does not provide quantitative metrics to evaluate the efficiency of these techniques. For the pre-trained data used in the experiments, ITIT still uses 4M paired data and 398M unpaired data to compete with the method that uses 398M paired data. In that case, the overall computational overhead is larger than the traditional paired data training. Can you provide some analysis in comparing the training efficiency between the paired data training and the ITIT.

2. The pseudo-synthesized data pairs could be noisy at the beginning of the training since the generator is not trained yet, I am wondering whether the pseudo-data pairs generated at the beginning will play negatively on the performance.

**Questions:**

1. Question for the gradient estimation of the cycle training: Referring to the figure 3, T2I is performed twice with a stop gradient operation in the middle for the T2I2T, why not directly perform the T2I once. Similar questions for the  twice I2T in the process of I2T2I.

2. Why ViT is chosen as the joint image-text encoder, instead of using some multimodal architectures. How does ViT extract the text feature?

---

> ### Author Response · Authors · 2023-11-21
> **Response to Reviewer NW6v**
>
> We thank the reviewer for the positive feedback and noting that our method is innovative. Below we respond to individual comments.
>
> **Q1: Training efficiency**
>
> **A1:** We include a detailed training speed analysis in Table 2 of the revised Appendix (new texts in purple). ITIT is ~2.32x slower than the no-cycle baseline during training (but the same speed as the baseline during inference). However, we note that the cost of doubling the training time is marginal compared to the cost of collecting large-scale paired datasets. We show in our paper that ITIT trained with only 4M paired data can achieve a similar performance as baseline trained with 400M paired data, reducing the cost of collecting paired data by 100 times.
>
> **Q2: Training schedule**
>
> **A2:** This is an interesting suggestion by the reviewer. We experimented with the two-stage training scheme with ITIT-B trained for 500K steps, starting the second stage training (cycle loss) at different training steps. As shown in Table 3 in the revised Appendix (new text in purple), starting the second stage at 25K steps achieves slightly better performance than starting from the beginning. Interestingly, introducing the cycle loss at the very beginning (0 steps) does not significantly impair performance. This resilience is likely due to the dominance of paired data in the early training stages. As the generation results improve, the impact of cycle consistency loss becomes more pronounced, achieving greater diversity and generalization ability through the utilization of unpaired data.
>
> **Q3: Why performing T2I twice in T2I2T cycle**
>
> **A3:** In ITIT, the synthesized image and text need to be of reasonable quality to help the cycle training. However, existing T2I and I2T methods cannot generate high-quality images or texts in one step. Therefore, we need to first generate the images and texts in multiple steps (24 steps for T2I and auto-regressively for I2T) to ensure it is of reasonable quality, and then back-propagate the cycle-consistency loss to one of these generation steps.
>
> **Q4: Why using ViT for image-text encoder**
>
> **A4:** Before the image-text encoder, we use a pre-trained T5 encoder to extract text embeddings from raw text tokens. The text embeddings are then concatenated with the image token embeddings as the input to our image-text encoder. Conceptually, our image-text encoder could use any multimodal network architecture. For simplicity, we use a simple architecture consisting of a stack of Transformer blocks, where each block consists of a multi-head self-attention block and an MLP block. We call it a “ViT” because it takes in both visual and text inputs, but it uses a general architecture that is not specific to visual inputs. We will clarify this term in the revised manuscript.

---

### Official Review · Reviewer_ySUK · 2023-10-29

**Soundness:** 3 good
**Presentation:** 3 good
**Contribution:** 3 good
**Rating:** 6
**Confidence:** 4

**Summary:**

The ITIT model introduces a new training paradigm in vision-language generative models, focusing on cycle consistency to facilitate training on unpaired image and text data. Unlike conventional models reliant on vast, paired image-text datasets, ITIT optimizes performance using a limited set of paired data, supplementing it with larger unpaired datasets, and ensures output accuracy through bidirectional image-to-text and text-to-image generation. Experimental results show that ITIT, even with significantly fewer paired datasets, delivers performance comparable to state-of-the-art models in image generation and captioning.

**Strengths:**

This paper proposes a view of training of image-to-text and text-to-image models using unpaired image and text data. The overall design is based on an existing reasonable idea of "cycle consistency".

The paper evaluates the proposed ITIT framework and the image-text cycle consistency method and demonstrates that it can enhance model performance in image generation and captioning.

The proposed method is claimed to be scalable to get higher performance improvement.

The paper writing is clear and easy to follow.

**Weaknesses:**

Innovation. It seems all innovation comes down to how to reduce memory consumption. Others such as network structures and designs already exist. Specifically, regarding the method of reducing memory consumption, the core idea is that the author found that generating training data from 0 to 1 has too high a memory cost, so they first go from 0 to 1 (without calculating the gradient), then manually return to 0.9, and redo 0.9 to 1. From this point of view, the contribution is somewhat engineering-oriented.

Other issues. There are a few other issues:
1. The author only mentions a reduction in memory usage, but the time consumption should be quite large, which is a disadvantage. Any justification for this?
2. The I2T2I method seems a bit unreasonable because the latter half of T2I can already generate many kinds of images, so why must reconstructing the image necessarily be correct?
3. For visual-language cycle consistency, there are two papers already: 'Equivariant Similarity for Vision-Language Foundation Models' (ICCV 2023), and 'CyCLIP: Cyclic Contrastive Language-Image Pretraining' (NeurIPS 2022). In this submission, there is an image-text encoder, for which it’s very straightforward to use the above two methods as a baseline (in the cycle consistency of embedding). So the reviewer is quite curious about how the cycle of embedding compares to the cycle of generating data.

**Questions:**

See the three questions in weaknesses "other issues".

---

> ### Author Response · Authors · 2023-11-21
> **Response to Reviewer ySUK**
>
> We thank the reviewer for the feedback. We respectfully disagree about innovation being a weakness of our method. As reviewers xcv8, NW6v, and Qb8x all point out, ITIT is a novel method that explores how to leverage unpaired data in vision-language generative pre-training. The core contribution of ITIT is to significantly reduce the need for paired data and maintain the performance with unpaired data via cycle consistency. Below we respond to individual comments. We hope the reviewer will consider our responses to their and other reviewers’ comments and reconsider their rating.
>
> **Q1: Training time**
>
> **A1:** We include a detailed training speed analysis in Table 2 of the revised Appendix (new texts in purple). ITIT is ~2.32x slower than the no-cycle baseline during training (but the same speed as the baseline during inference). However, the cost of doubling the training time is acceptable compared to the huge cost of collecting large-scale high-quality paired datasets. We show in our paper that ITIT trained with only 4M paired data can achieve a similar performance as baseline trained with 400M paired data, reducing the cost of collecting paired data by 100 times.
>
> **Q2: Why must reconstructing the image**
>
> **A2:** Our I2T2I reconstruction loss is a cross-entropy loss between reconstructed image tokens and ground-truth image tokens. Such a cross-entropy-based reconstruction loss is widely used in many prior image generation works [1, 2, 3]. Instead of reconstructing a single image as the reviewer’s question implied, the cross-entropy loss asks the model to maximize the log-likelihood of the input image. For example, multiple dog images could be mapped to the same text “dog”. The T2I model is then asked to reconstruct the image tokens from the text prompt “dog” so that it maximizes the log-likelihood of all the dog images.
>
> A similar cross-entropy-based reconstruction loss is also widely employed in many other tasks. One of them is training large language models. During training, the network is trained to predict the next token in the ground-truth sentence given the current tokens, which could have multiple possibilities. The loss is a cross-entropy loss between the reconstructed text token and the ground-truth text token, which helps the LLM to maximize the log-likelihood of the next token.
>
> **Q3: Prior works in visual-language cycle consistency**
>
> **A3:** We appreciate the reviewer's effort in highlighting relevant literature. Unlike these papers, which concentrate on visual representation learning, ITIT is dedicated to image-to-text (I2T) and text-to-image (T2I) generation tasks. Furthermore, while these papers utilize text supervision to enhance visual representations through image-text similarity regularizations, they rely on **paired** image-text data for this purpose, as evident in Equation (6) of the ICCV 2023 paper [6]. In contrast, ITIT employs cycle consistency to leverage **unpaired** data for vision-language generative pre-training.
>
> We have also incorporated a discussion of these papers in the related work section of our revised manuscript (new texts in purple). It's important to note that our current image-text encoder is not pre-trained using any image-text representation learning methods, ensuring a fair comparison with prior works [1, 2]. An intriguing future research would be to explore the use of an encoder pre-trained with methods such as CLIP [4] as well as the papers suggested by the reviewer ([5, 6]) as the initialization for our image-text encoder.
>
> **References**
>
> [1] Huiwen Chang, et al. Muse: Text-to-image generation via masked generative transformers. arXiv preprint arXiv:2301.00704, 2023.
>
> [2] Lili Yu, Bowen Shi, Ramakanth Pasunuru, Benjamin Muller, Olga Golovneva, Tianlu Wang, Arun Babu, Binh Tang, Brian Karrer, Shelly Sheynin, et al. Scaling autoregressive multi-modal models: Pretraining and
> instruction tuning. arXiv preprint arXiv:2309.02591, 2023.
>
> [3] Patrick Esser, Robin Rombach, and Bjorn Ommer. Taming transformers for high-resolution image synthesis. In Proceedings of the IEEE/CVF conference on computer vision and pattern recognition, pp. 12873–12883, 2021.
>
> [4] Alec Radford, Jong Wook Kim, Chris Hallacy, Aditya Ramesh, Gabriel Goh, Sandhini Agarwal, Girish Sastry, Amanda Askell, Pamela Mishkin, Jack Clark, et al. Learning transferable visual models from natural language supervision. In International conference on machine learning, pp. 8748–8763. PMLR, 2021.|
>
> [5] Goel, Shashank, Hritik Bansal, Sumit Bhatia, Ryan Rossi, Vishwa Vinay, and Aditya Grover. "Cyclip: Cyclic contrastive language-image pretraining." Advances in Neural Information Processing Systems 35 (2022): 6704-6719.
>
> [6] Wang, Tan, Kevin Lin, Linjie Li, Chung-Ching Lin, Zhengyuan Yang, Hanwang Zhang, Zicheng Liu, and Lijuan Wang. "Equivariant Similarity for Vision-Language Foundation Models." arXiv preprint arXiv:2303.14465 (2023).

---

> ### Author Response · Authors · 2023-11-22
>
> Dear Reviewer ySUK,
>
> We sincerely appreciate your valuable feedback on our submission. As the author-reviewer discussion nears its end, we are eager to know if our responses have addressed your concerns. We are also more than willing to engage in further discussions if needed.
>
> Thank you again for your time and effort!

---

### Official Review · Reviewer_Qb8X · 2023-11-01

**Soundness:** 3 good
**Presentation:** 3 good
**Contribution:** 3 good
**Rating:** 6
**Confidence:** 3

**Summary:**

This paper suggests novel  ITIT(InTegrating Image Text) method which enables leveraging massive unpaired images or texts. It suggests a concept of cycle consistency by using image-to-text (I2T) generation model and text-to-image (T2I) generation model in a single framework. As a result, the authors verify that ITIT with unpaired data shows comparable performance with paired data and, it achieves state-of-the-art performance with much fewer paired image-text data.

**Strengths:**

1. The motivation (usage of unpaired image and text) is clear and solid.
2. The method (ITIT) is a simple but strong:
     - It introduces the concept of consistency, effectively bridging image-to-text generation models (like BLIP) with text-to-image generation models (like Maze).1
     - The two-step phase in T2I2T / I2T2I facilitates efficient pre-training.
3. The experimental result is strong. To the best of my knowledge, this might be the first paper where a method pre-trained on unpaired image-text data demonstrates comparative performance with those pre-trained on paired image-text datasets.
4. The paper is well-written and figure and tables are well-organized.

**Weaknesses:**

1. In Table 2, the gap between full cycle and half cycle of ITIT training seems very marginal, which raises doubts about the effectiveness of proposed consistency concept. Namely, it seems that the pseudo-targets (ITIT-half cycle) would be sufficient to leverage unpaired image and texts. In this case, I believe that generating pseudo-targets would be more powerful:  1) it is computationally efficient and 2) it could easily integrate with the recent pre-trained models such as BLIP v2, stable-diffusion, and so on. (since the generation process of pseudo-targets are model-agnostic)


2. Given that ITIT employs synthetically generated captions or images during the training process, there is a reasonable expectation that numerous failure cases may occur. This concern is particularly pronounced with datasets like CC3M, which are known to consist of noisy image-text pairs, potentially leading to a higher frequency of failures.
 It would be grateful if authors could share the fair cases.

**Questions:**

1. Could you provide a comparison of the computational overhead involved in ITIT when implemented in full cycle, half cycle, and no-cycle
2. Are the models involved in this study trained entirely from scratch, or do they utilize weights from pre-existing pre-trained models?
3. Is it possible to extend the method for using diffusion based model for the T2I?
4. Given the concerns mentioned in the weakness section about ITIT's susceptibility to failure cases in the early pre-training phase, I wonder:  What are the results when trained with a two-stage approach  (first pre-trained with paired dataset and then use unpaired dataset / paired-unpaired dataset)?

---

> ### Author Response · Authors · 2023-11-21
> **Response to Reviewer Qb8X (1/2)**
>
> We thank the reviewer for the insightful feedback. We are happy that the reviewer recognizes the novelty of our work. We respond to individual comments below. We hope the reviewer would consider our responses to their and other reviewers’ comments and reconsider their rating.
>
> **Q1: Half-cycle ITIT vs. full-cycle ITIT**
>
> **A1:** We are grateful for the reviewer's recognition of the effectiveness of our half-cycle training in achieving notable performance. Indeed, the core objective of the ITIT framework is to harness unpaired image and text data for vision-language generative training. Through our experiments, we demonstrate that both half-cycle and full-cycle iterations of ITIT are capable of leveraging unpaired data to significantly enhance image-to-text (I2T) and text-to-image (T2I) generation.
>
> As detailed in Table 2 of the main paper and Table 2 of the revised appendix (new texts in purple), it is evident that full-cycle training offers additional improvements over half-cycle training. Notably, this enhancement comes with only a marginal increase in training speed overhead (around 10%) and no additional inference overhead. The performance of full-cycle training can be attributed to its more rigorous enforcement of cycle consistency, as illustrated in Figure 6 of the main paper.
>
> Conversely, we acknowledge that implementing half-cycle training with other pre-training methods may present a more straightforward approach. This consideration reflects a trade-off inherent in the two versions of ITIT: while full-cycle training offers higher performance due to its comprehensive cycle consistency, half-cycle training provides greater flexibility and ease of adaptation to various pre-training methodologies. This balance between flexibility and performance efficiency is a key aspect of our ITIT framework's design and application.
>
> **Q2: Failure cases**
>
> **A2:** We thank the reviewer for the suggestion. We added some failure cases in Fig. 5 (page 9)  of the revised Appendix. As shown in the figure, ITIT encounters failure cases that are common across most image-to-text generation models. Some of the issues, such as watermarks, are closely tied to the quality of the data utilized in training. Given this context, we believe that ITIT has the potential to mitigate these prevalent shortcomings by lessening the dependency on paired data in vision-language generative training. A promising direction for future research would involve training ITIT with a combination of a small but exceptionally high-quality, paired image-text dataset and extensive, high-quality unpaired image and text datasets.
>
> **Q3: Computational overhead**
>
> **A3:** In the revised appendix of our paper (new texts in purple), we provide an in-depth analysis of training speeds, as presented in Table 2. Our half-cycle ITIT framework is 2.08 times slower than the no-cycle baseline during training, and our full-cycle ITIT framework is 2.32 times slower, while incurring no additional time expenditure during the generation phase. It's important to emphasize, however, that this increase in training time is relatively minor when compared to the extensive resources required for compiling large-scale paired datasets.
>
> **Q4: Training from scratch**
>
> **A4:** As mentioned in Appendix B, similar to prior works [1, 2], we use a pre-trained T5 encoder to extract text embeddings from raw texts and use a pre-trained VQGAN tokenizer to tokenize raw images into image tokens. The T5 encoder and VQGAN tokenizer are fixed during the training of ITIT. All other modules, including our image-text encoder, text decoder, and image decoder are trained from scratch. Note that T5 and VQGAN are trained on a single modality, text or image, respectively.

---

> ### Author Response · Authors · 2023-11-21
> **Response to Reviewer Qb8X (2/2)**
>
> **Q5: Extend to diffusion-based model**
>
> **A5:** ITIT is a general vision-language generative training scheme that could potentially be adapted to different T2I frameworks, including a diffusion-based model. However, since the ITIT framework needs to perform T2I generation in an online manner, one important property we consider when choosing our T2I framework is the inference speed. Thus, In this paper, we adopt a parallel decoding T2I framework similar to Muse [1], which is shown to be very efficient for T2I generation. Other frameworks with fast inference speed, such as latent diffusion [3] or latent consistency model [4], could be also applied in ITIT by simply replacing the T2I module with the diffusion-based model. For example, an intuitive way to incorporate the diffusion model in cycle training is to use a similar strategy as ITIT, which first generates the image with multiple diffusion sampling steps, and then back-propagates the gradient to one of the diffusion steps for cycle training. However, we note that the primary aim of our paper is to demonstrate the potential and advantages of incorporating unpaired data into vision-language training. The integration of other text-to-image (T2I) and image-to-text (I2T) frameworks, while equally important, is designated for future research endeavors.
>
> **Q6: Two-stage training**
>
> **A6:** We thank the reviewer for the suggestion. We experimented with the two-stage training scheme with ITIT-B trained for 500K steps, starting the second stage training (cycle loss) at different training steps. As shown in Table 3 in the revised Appendix, starting the second stage at 25K steps achieves slightly better performance than starting from the beginning. Interestingly, introducing the cycle loss at the very beginning (0 steps) does not significantly impair performance. This resilience is likely due to the dominance of paired data in the early training stages. As the generation results improve, the impact of cycle consistency loss becomes more pronounced, achieving greater diversity and generalization ability through the utilization of unpaired data.
>
> **References**
>
> [1] Huiwen Chang, Han Zhang, Jarred Barber, AJ Maschinot, Jose Lezama, Lu Jiang, Ming-Hsuan Yang, Kevin Murphy, William T Freeman, Michael Rubinstein, et al. Muse: Text-to-image generation via masked generative transformers. arXiv preprint arXiv:2301.00704, 2023.
>
> [2] Yu, Lili, Bowen Shi, Ramakanth Pasunuru, Benjamin Muller, Olga Golovneva, Tianlu Wang, Arun Babu et al. "Scaling autoregressive multi-modal models: Pretraining and instruction tuning." arXiv preprint arXiv:2309.02591 (2023)
>
> [3] Robin Rombach, Andreas Blattmann, Dominik Lorenz, Patrick Esser, and Bjorn Ommer. High-resolution ¨ image synthesis with latent diffusion models. In Proceedings of the IEEE/CVF Conference on Computer Vision and Pattern Recognition, pp. 10684–10695, 2022.
>
> [4] Luo, Simian, Yiqin Tan, Longbo Huang, Jian Li, and Hang Zhao. "Latent consistency models: Synthesizing high-resolution images with few-step inference." arXiv preprint arXiv:2310.04378 (2023)

---

> ### Author Response · Authors · 2023-11-22
>
> Dear Reviewer Qb8X,
>
> We sincerely appreciate your valuable feedback on our submission. As the author-reviewer discussion nears its end, we are eager to know if our responses have addressed your concerns. We are also more than willing to engage in further discussions if needed.
>
> Thank you again for your time and effort!

---

> > ### Comment · Reviewer_Qb8X · 2023-11-22
> >
> > I appreciate the detailed reply. I have carefully read them and choose to keep my original score.

---

### Official Review · Reviewer_xcv8 · 2023-11-04

**Soundness:** 3 good
**Presentation:** 4 excellent
**Contribution:** 3 good
**Rating:** 8
**Confidence:** 4

**Summary:**

This paper proposes to learn from unpaired data for T2I I2T generation by exploiting a cycle consistency constraint. A joint text-image architecture is presented as the unify model for this learning. Then, pseudo examples are generated and cycle consistency is optimized as two sub-steps in one training iteration. The learned model produces good generation quality with limited paired data, such as 4M vs. 398M.

**Strengths:**

1. The semi-supervised learning process for generative models has not been well studied before.
2. This paper provides a plausible way of learning from unpaired data with optimized computation cost.
3. Ablation studies are well presented and support the claim of adopting unpaired data.

**Weaknesses:**

1. This paper claims that the paired data gap is almost closed in Figure 4 (from 398M to 4M). However, there is still a significant gap in FID and CIDEr according to Figure 1 compared with other methods e.g. CM3Leon, Muse, GIT. Is there any other reason besides different numbers of parameters? Or maybe Shutterstock data is so similar that scaling data does not make a difference, but on more diverse datasets the Figure 4 claim will not hold?
2. How does ITIT generalize to new/different domains without paired data in that domain (assuming unpaired domain specific text and image available)? One example readers might be interested in could be medical imaging where paired data is much more difficult to obtain.
3. Unclear compatibility with other types of generative models such as diffusion/flow/consistency models as the I2T stage. How much does the ITIT framework depends on the unified architecture? Does ITIT work with any kind of T2I I2T model?
4. ITIT essentially denoises training data by learning a better alignment as shown in Figure 6. Does it worth to study other methods that also cleans the data, e.g. simply CLIP filtering with high confidence as pseudo labels, i.e. semi-supervised discriminative learning? This might keep the “blue” information in Figure 6 as well?
5. Runtime due to the pseudo pair generation stage. The appendix mentioned that the ITIT requires 2x training time compared with T2I and I2T non-cycle training, but how is it the case considering the 24 steps of decoding happening? How much of the time was spent on the online pseudo pair generation stage? Does it require careful batching or implementation for good TPU utilization?
6. Although scaling data is left to future work, the value of this paper depends a lot on the promise of further scaling unpaired image and text data beyond the combination of all current paired datasets. Are we still able to collect 10x high quality images/texts but not paired? Do we still need more data in this domain? Unfortunately this has not been shown to a good extent.

**Questions:**

(discussed in the weakness section)

---

> ### Author Response · Authors · 2023-11-21
> **Response to Reviewer xcv8 (1/2)**
>
> We thank the reviewer for the positive comments and inspiring feedback. We are glad that the reviewer noted the novelty of the task we studied. We respond to individual comments below:
>
> **Q1: Comparison with other methods.**
>
> **A1**: We acknowledge the reviewer's point regarding the comparison between ITIT and existing text-to-image (T2I) and image-to-text (I2T) methods, such as CM3Leon, Muse-512, and GIT. It is important to note that these methods are trained with significantly more parameters (3B for Muse and 7B for CM3Leon) and paired data (860M for Muse and 360M for CM3Leon) than our ITIT model. GIT is also trained on a substantially larger dataset (800M), employs a pre-trained CLIP model for initializing its image encoder (as opposed to training from scratch), and incorporates the COCO training dataset in its training regime, the test set of which is used for evaluation. Therefore, drawing a direct performance comparison between ITIT and these methods would be inequitable.
>
> In Table 1 of our paper, we present the results of training GIT and Muse under conditions more comparable to those of ITIT. Under similar constraints of using the same paired dataset (CC3M) and a comparable number of network parameters, ITIT demonstrates superior performance in terms of both FID and CIDEr scores. This improvement is attributed to the effective leveraging of unpaired data from WebLI, facilitated by the cycle consistency loss.
>
> It's crucial to point out that the experiments in Table 1 and Figure 4 utilize different datasets. Figure 4 employs Shutterstock for both paired and unpaired data, whereas the results in Table 1 are derived from using CC3M as the paired data source and WebLI as the unpaired one. Given that CC3M exhibits the highest image-text pairing quality among the datasets we considered, ITIT achieves better performance on the CC3M+WebLI combination compared to Shutterstock.
>
> The primary focus of Figure 4 is to investigate how ITIT enhances performance over the baseline using the same amount of paired data augmented with additional unpaired data. To effectively study this, the added unpaired data should not present a domain gap with the paired data, rendering the CC3M+WebLI setting impractical for this specific analysis. Therefore, we opted to conduct this part of the experiment using Shutterstock. This choice also demonstrates that the cycle consistency mechanism is beneficial across different datasets, further highlighting its versatility and effectiveness.
>
> **Q2: How does ITIT generalize to new/different domains without paired data in that domain?**
>
> **A2:** We believe that our ITIT model has the potential to significantly reduce the reliance on extensive paired data in specialized fields like medical imaging. This reduction in dependence on paired data could open up new possibilities for domain adaptation, where paired data from one domain might facilitate effective model training using unpaired data from other domains. Exploring this possibility of cross-domain adaptation using ITIT represents an exciting and promising avenue for future research.
>
> **Q3: Does ITIT work with any kind of T2I I2T model?**
>
> **A3:** ITIT represents a general vision-language generative training scheme, adaptable to various text-to-image (T2I) and image-to-text (I2T) frameworks. The core principle of employing cycle consistency loss to effectively utilize unpaired image and text data is universally applicable. Nonetheless, an essential aspect of the ITIT framework is its ability to perform T2I generation in an online manner, necessitating a focus on inference speed when selecting a suitable T2I framework.
>
> In our study, we choose a parallel decoding T2I framework similar to Muse [1], known for its efficiency in T2I generation. Our proposed unified image-text encoder further emphasizes simplicity and reduced training costs. It's important to clarify that these attributes are not obligatory for the implementation of ITIT. The framework is flexible enough to accommodate other fast inference models, such as latent diffusion [2] or latent consistency models [3]. However, we note that the primary aim of our paper is to demonstrate the potential and advantages of incorporating unpaired data into vision-language training. The integration of other text-to-image (T2I) and image-to-text (I2T) frameworks, while equally important, is designated for future research endeavors.

---

> ### Author Response · Authors · 2023-11-21
> **Response to Reviewer xcv8 (2/2)**
>
> **Q4: Other methods that also clean the data.**
>
> **A4:** The primary objective of our ITIT framework is to effectively utilize unpaired image and text data within the context of vision-language generative training. In contrast, existing data cleaning methods, like CLIP filtering, are designed to refine noisy paired datasets, enhancing the quality of image-text pairings. These methods, however, are not suited for handling unpaired data. Consequently, ITIT should be seen as complementary to data filtering approaches. While the latter focuses on elevating the quality of paired data, ITIT is designed to harness the potential of unpaired data. This distinction underscores ITIT's unique contribution to the field, offering a novel approach that broadens the scope of vision-language training beyond the confines of traditional paired data methodologies.
>
> **Q5: Training cost.**
>
> **A5:** In the revised appendix of our paper (new texts in purple), we have included a comprehensive training cost analysis, presented in Table 2. In the scenario of T2I2T half-cycle training (row 2), we observe that the speed is 1.4 times slower than that of the no-cycle approach (row 1). Specifically, for row 1, a batch size of 1024 is allocated for T2I training, while the remaining 1024 is used for I2T/I2I training. In contrast, row 2 involves 512 data for T2I, 512 for I2T/I2I, and an additional 1024 dedicated to T2I2T training.
>
> The primary reason for this increased training time is the 24-step T2I online generation process. However, it's important to note that a significant portion of the training time is consumed by the gradient backpropagation step, which is the same for no-cycle and half-cycle cycle training and slightly higher for full-cylce training. Consequently, the additional time required for the online generation phase is considered acceptable within the overall training process.
>
> **Q6: Data scaling in vision-language generative training.**
>
> **A6:** This is a great question which we believe should be investigated further. As shown in many recent works, the model and data size of vision-language generative models are far from saturated, and the quality of image-text pairing is also crucial in generation performance [4, 5, 6, 7]. However, paired data is much harder to collect than unpaired data. Pali [5], for example, shows that they are able to collect 10 billion images and 12 billion texts from the internet, but after image-text alignment filtering, only 10% of the image-text pairs are kept. Therefore, ITIT has great potential in utilizing unpaired image and text data to improve the performance of vision-language generative models. Moreover, we believe that the idea of utilizing unpaired data in multimodal generative training could be broadly applied to other modalities where collecting paired data is even harder, such as medical imaging or disease diagnosis.
>
> **References**
>
> [1] Huiwen Chang, Han Zhang, Jarred Barber, AJ Maschinot, Jose Lezama, Lu Jiang, Ming-Hsuan Yang, Kevin Murphy, William T Freeman, Michael Rubinstein, et al. Muse: Text-to-image generation via masked generative transformers. arXiv preprint arXiv:2301.00704, 2023.
>
> [2] Robin Rombach, Andreas Blattmann, Dominik Lorenz, Patrick Esser, and Bjorn Ommer. High-resolution ¨ image synthesis with latent diffusion models. In Proceedings of the IEEE/CVF Conference on Computer Vision and Pattern Recognition, pp. 10684–10695, 2022.
>
> [3] Luo, Simian, Yiqin Tan, Longbo Huang, Jian Li, and Hang Zhao. "Latent consistency models: Synthesizing high-resolution images with few-step inference." arXiv preprint arXiv:2310.04378 (2023)
>
> [4] Betker et. al., Improving Image Generation with Better Captions, 2023
>
> [5] Xi Chen, Xiao Wang, Soravit Changpinyo, AJ Piergiovanni, Piotr Padlewski, Daniel Salz, Sebastian Goodman, Adam Grycner, Basil Mustafa, Lucas Beyer, Alexander Kolesnikov, Joan Puigcerver, Nan Ding, Keran Rong, Hassan Akbari, Gaurav Mishra, Linting Xue, Ashish Thapliyal, James Bradbury, Weicheng Kuo, Mojtaba Seyedhosseini, Chao Jia, Burcu Karagol Ayan, Carlos Riquelme, Andreas Steiner, Anelia Angelova, Xiaohua Zhai, Neil Houlsby, and Radu Soricut. Pali: A jointly-scaled multilingual language-image model, 2023.
>
> [6] Jiahui Yu, Zirui Wang, Vijay Vasudevan, Legg Yeung, Mojtaba Seyedhosseini, and Yonghui Wu. Coca: Contrastive captioners are image-text foundation models. arXiv preprint arXiv:2205.01917, 2022a.
>
> [7] Jiahui Yu, Yuanzhong Xu, Jing Yu Koh, Thang Luong, Gunjan Baid, Zirui Wang, Vijay Vasudevan, Alexander Ku, Yinfei Yang, Burcu Karagol Ayan, et al. Scaling autoregressive models for content-rich text-to-image generation. arXiv preprint arXiv:2206.10789, 2(3):5, 2022b.

---

### Meta-Review · Area_Chair_b1Zv · 2023-12-06

**Metareview:**

This paper proposes ITIT (InTegrating ImageText), a novel cycle consistency based training paradigm that enables vision-language training on unpaired image and text data. The paper received unanimous accept recommendations (8, 6, 6, 8) from reviewers. The paper's strengths include its novelty of cycle consistency based generative model training on unpaired image and text, the model scaling well with unpaired data, strong experimental results and thorough ablation studies, and clear exposition. Most of the initial weaknesses surrounding discussion of related work, paper claims, questions regarding generalizability to different domains, compatibility with other types of generative models, and gap between full cycle and half cycle, and questions regarding efficiency were adequately addressed by the rebuttal. The ACs agree with the reviewers that this paper can make a strong contribution to the machine learning and computer vision communities; in particular, the proposed approach is simple and effective, and presents an innovative idea. The ACs thus recommend spotlight acceptance.

**Justification For Why Not Higher Score:**

Based on the strengths written in the meta review, this paper deserves a spotlight presentation. It makes an interesting and solid contribution to the area of vision-language generative models.

**Justification For Why Not Lower Score:**

All reviewers and ACs advocate for acceptance of this paper.

---

### Decision · Program_Chairs · 2024-01-16

Accept (spotlight)